# Biological, Psychiatric, Psychosocial, and Cognitive Factors of Poststroke Depression

**DOI:** 10.3390/ijerph20075328

**Published:** 2023-03-30

**Authors:** Mario F. Dulay, Amber Criswell, Timea M. Hodics

**Affiliations:** 1Houston Methodist Neurological Institute, 6560 Fannin Suite 944, Houston, TX 77030, USA; 2Department of Neurosurgery, Houston Methodist Hospital, Houston, TX 77030, USA; 3Department of Neurology and Eddy Scurlock Stroke Center, Houston Methodist Hospital, Houston, TX 77030, USA

**Keywords:** cerebrovascular accident, poststroke depression, poststroke anxiety, memory loss

## Abstract

Background: Depression is the most common psychiatric condition that occurs after cerebrovascular accident, especially within the first year after stroke. Poststroke depression (PSD) may occur due to environmental factors such as functional limitations in daily activities, lower quality of life, or biological factors such as damage to areas in the brain involved in emotion regulation. Although many factors are hypothesized to increase the risk of PSD, the relative contribution of these factors is not well understood. Purpose: We evaluated which cross-sectional variables were associated with increased odds of PSD in our adult outpatient stroke neuropsychology clinic population. Methods: The sample included 325 patients (49.2% female; mean age of 59-years old) evaluated at an average of 8.1 months after an ischemic or hemorrhagic stroke. Variables included in logistic regression were stroke characteristics, demographics, psychosocial factors, comorbid medical problems, comorbid psychiatric conditions, and cognitive status. The Mini International Neuropsychiatric Inventory was used to determine DSM-defined PSD and anxiety disorders. A standard neuropsychological test battery was administered. Results: PSD occurred in 30.8% of the sample. Logistic regression indicated that increased odds of PSD were associated with a comorbid anxiety disorder (5.9 times more likely to suffer from PSD, *p* < 0.001). Further, increased odds of PSD were associated with a history of depression treatment before stroke (3.0 times more likely to suffer from PSD), fatigue (2.8 times more likely), memory impairment (2.4 times more likely), and younger age at stroke (all *p* values < 0.006). Discussion: Results suggest that PSD is likely multifactorial and extends the literature by demonstrating that a comorbid anxiety disorder correlated strongest with PSD. Poststroke screening and treatment plans should address not only depression but comorbid anxiety.

## 1. Introduction

Poststroke depression (PSD) is common in a subset of individuals after stroke. Literature reviews have found that between 18% and 33% of stroke patients develop depression after stroke [1,2] and this figure may be as high as 55% in individual studies [3]. A more severe level of depression is associated with greater disability, greater cognitive impairment, increased mortality, and reduced quality of life [4]. Yet, PSD continues to be under-reported, under-identified, and undertreated.

Significant variability exists in the reported rates of PSD given that research studies use different instruments to estimate depression. A limitation with previous research was that depression was defined by self-report screeners, such as the Beck Depression Inventory and Patient Health Questionnaire-9, that quantify the presence of depressive symptoms but do not determine whether or not the symptoms affect a person’s daily life. Screening measures of depression are important in identifying and triaging patients who may be suffering from PSD, but follow-up should occur for a more definitive diagnosis of clinical depression [5]. Several studies have used the Diagnostic and Statistical Manual of Mental Disorders (DSM) standard that poststroke depressive symptoms should cause clinically significant impairment in social, academic, or occupational functioning in order to be considered a depressive disorder, but, problematically, most studies that used DSM criteria had small sample sizes when conducting regression analyses, limiting the number of risk factors that could be simultaneously studied [6].

The development of depression after stroke is multifactorial and complex. Previous studies identified several risk factors for PSD. For example, PSD occurs at a greater rate after left-hemispheric stroke [7], strokes in the frontal lobe or basal ganglia [8], more anterior location of stroke [9], more severe stroke [3], level of physical disability [10], level of dependence on others [11], history of previous stroke [11], chronic accumulation of small macrovascular and microvascular lesions [12], female gender [13], treatment for depression prior to stroke [10], diabetes mellitus [14], history of myocardial infarction [15], stressful life events in the months before stroke [13], living alone at time of stroke [16], Medicaid patients (versus private insurance patients) [15], and lower educational attainment [16]. Comorbidities such as pseudobulbar affect [13], anxiety [3], and greater cognitive impairment are more common in patients with PSD [17]. Despite these findings, there are also many studies that show conflicting results [18]. Recently, using a chart review of archival data to estimate depression and anxiety, a history of an unspecified type of anxiety was found to be associated with PSD [19]. Although many factors may be associated with the increased risk of PSD, the relative contribution of these factors is not well understood. Knowing the risk factors and comorbidities of PSD will help clinicians target patients for referral to pharmacologic and behavioral interventions to improve functional recovery and outcome.

The current study evaluated cross-sectional factors for DSM-defined PSD in a large sample from our neuropsychology clinic at Houston Methodist Hospital. Associated factors included stroke-related, demographic, psychosocial, comorbid medical, history of psychiatric treatment, and cognitive issues.

## 2. Materials & Methods

### 2.1. Sample

Three-hundred and twenty-five patients who had sustained an ischemic or hemorrhagic stroke were evaluated at an average of 8.3 months after stroke in this prospective IRB-approved single center study. Evaluations only occurred at the Houston Methodist Hospital in quiet dedicated clinic rooms. Patients presented from diverse settings including home, retirement communities, inpatient rehabilitation facilities, and skilled-nursing facilities.

### 2.2. Inclusion Criteria

Completed neuropsychological and psychological assessment. Participants who were evaluated by the Neurosurgery department’s Neuropsychology Service of the Houston Methodist Neurological Institute at the Houston Methodist Hospital between 1 July 2009 and 6 September 2019.Verified stroke: Location and lateralization of strokes were defined by neuroimaging neuroradiologist clinical reports. Lateralization of strokes included 44.9% left hemisphere, 41.2% right hemisphere, and 13.8% bilateral. Locations of strokes were as follows: 35.7% MCA, 23.1% frontal lobe/ACA stroke, 14.8% cerebellar, 9.9% basal ganglia, 6.5% pontine, 4.6% thalamic, 3.4% PCA, 3.4% occipital, and 1.8% parietal. Stroke types included 65.1% ischemic and 31.9% hemorrhagic.

### 2.3. Exclusion Criteria

92 patients were excluded from the study for the following reasons:Under the age of 18-years-old.A Mini Mental Status Exam score below 20 or invalid testing. All evaluations were deemed valid or invalid before interpreting the results for clinical purposes by considering comprehension, effort, and language proficiency. Often, it was one neurocognitive test that was deemed invalid rather than the entire evaluation.When testing was deemed invalid due to sensory loss. Some patients had reduced hearing, and if their hearing devices were not deemed adequate, an additional hearing device (earphones and an amplifier) was provided. Patients were included with visual field cuts and visual neglect. No patients were excluded because of immobility. If patients could not use their upper extremities to write, non-motor neurocognitive tests were used to define memory or executive impairments.A stroke history on multiple occasions.Those with other non-stroke-related neurologic diagnoses that would affect PSD (epilepsy, TBI, dementia). Patients with a history of stroke and a diagnosis of Vascular dementia were also excluded.Unknown type of stroke (ischemic, hemorrhagic) based on the neuroradiology report and clinical chart review.

### 2.4. Procedures

Neurocognitive Tasks: Standardized and normalized pencil-and-paper tests were used to evaluate verbal and visual memory and executive functioning (e.g., reasoning). Impairments in memory (e.g., Repeatable Battery Neurocognitive Screening, Hopkins Verbal Learning Test-Revised and California Verbal Learning Test-II for verbal memory) or executive functioning (e.g., Trail Making Test part B, Wisconsin Card Sort Test, Delis–Kaplan Executive Functions tests) were made categorical for analytic purposes because different cognitive tasks were administered over the 10 years of data collection. All tests were used in the evaluation of individuals seen at Houston Methodist Hospital who had been referred by a physical medicine and rehabilitation specialist or vascular neurologist. Administration and interpretation of the tests followed standardized procedures [20]. Impairment of the memory or executive tasks were dichotomized as impaired or intact defined by a z-score < −1.32.

Depression and Anxiety Diagnoses: The standard Mini International Neuropsychiatric Inventory (MINI) for DSM-IV or -5 [21] was used to determine a diagnosis of poststroke depression (including the following DSM diagnoses that began after the stroke: major depressive disorder (MDD) single episode, adjustment disorder with depressed mood, MDD recurrent, persistent depressive disorder or dysthymia, none with bipolar disorder) and poststroke anxiety disorders (panic disorder and generalized anxiety disorder [GAD], post-traumatic stress disorder, social anxiety disorder, obsessive–compulsive disorder patients). Interview questions were asked to the patient and available family members or caregivers as to “what limits existed in activities of daily living and whether or not those limits were due to physical weaknesses versus psychiatric symptoms or both.”

Comorbid medical issues and social factors used in analyses: Employment status was quantified using a history questionnaire and clinical interview to determine whether or not someone had gainful part-time or full-time employment at the time of the stroke. Marital status was defined as married or in a long-term domestic partner relationship versus single, dating, divorced, or widow/widower. Fatigue was evaluated by a clinical interview determining whether or not the person had significant tiredness, i.e., decreased energy or fatigue affecting that person’s life. Severity of fatigue or tiredness was further estimated based on the Beck Depression Inventory-2 questions asking about tiredness or fatigue. Sleep difficulty was evaluated by clinical interview, medical chart review from the recent referring physician’s notes, and use of the Beck Depression Inventory-2. Sleep trouble was defined as the self-reported level of difficulty falling asleep, staying asleep, or getting up repeatedly throughout the night without being able to go back to sleep (too much sleep was not defined as sleep difficulty in this study). The history of treatment for depression was based on a clinical interview and medical chart review and included treatment with medication or talk therapy or psychiatric hospitalization before the stroke.

### 2.5. Statistical Analyses

A one-way ANOVA was used to evaluate between-group differences (depressed, not depressed) for the demographic continuous variables (chronological age, education level, time since stroke). A Fisher’s exact test was used to evaluate between-group differences (depressed, not depressed) for the following categorical variables:stroke type (ischemic hemorrhagic);gender;% employed;% married;% with sleep difficulties;% with fatigue;% with history of treatment for depression, anxiety;% current anxiety disorder;% memory problems;% executive difficulties.

A logistic regression analysis was computed to evaluate cross-sectional associations with poststroke depression. Using SPSS-28, Depression Status (depressed, not depressed) was entered in block 0 as the independent variable, and then in block 1, using the enter method, the following dependent variables were used as predictors simultaneously: time since stroke, chronological age, gender, location of stroke, lateralization of stroke, stroke type (hemorrhagic/ischemic), sleep difficulties, fatigue, employment status, marital status, history of treatment for a depressive disorder, history of an anxiety disorder, current anxiety disorder, current memory difficulties, and current executive function difficulties.

## 3. Results

Table 1 lists the frequencies and means for the sample’s demographic, stroke, psychiatric, psychosocial, and cognitive characteristics. Of the 325 patients, 30.8% were diagnosed with PSD (N = 100), and 16.9% (55 patients) were diagnosed with a poststroke anxiety disorder. Thirty-one percent of the sample had been treated for depression prior to the stroke, and 7.4% had been treated for an anxiety disorder before the stroke. More patients sustained an ischemic stroke (67.1%) than a hemorrhagic stroke (32.9%). Not depicted in the table, of the 55 patients in this study diagnosed with a poststroke anxiety disorder, 43.6% (N = 24) had been treated for an anxiety disorder prior to the stroke, and in nearly every case, the anxiety type was the same (e.g., the two patients with PTSD had reoccurrence of their longstanding PTSD after the strokes). New onset poststroke anxiety occurred in 56.4% of poststroke anxiety patients (N = 31).

Table 2 displays the subtypes of poststroke anxiety disorders for the sample. Most individuals did not experience poststroke anxiety (83.1% of sample). The most common form of poststroke anxiety disorder was panic attacks followed by GAD.

Table 3 displays continuous variables (chronological age, education level, time since stroke to neuropsych testing) divided by depression status (PSD, No depression). Significant between-group differences were found with the one-way ANOVA (F (1333] = 910.0, *p* = 0.002) for chronological age such that patients who were depressed were younger. There were no significant differences for the level of educational attainment or time since stroke. Table 3 also displays all of the categorical variables’ percentages divided by depression status (PSD, No depression). Fisher’s exact test indicated significant between-group differences such that patients with PSD were more likely to have a history of depression treatment before stroke (61% of PSD patients compared to 28% of patients without depression), as well as experience difficulties at a higher frequency for stroke, type, sleep, fatigue, anxiety treatment before stroke, comorbid current poststroke anxiety disorder, memory problems, and executive difficulties (all *p* < 0.05 with executive functioning at *p* = 0.52). Fisher’s exact test analyses indicated no between-group differences based on depression status (PSD, No depression) for side-of-stroke, sex, employment status, or marital status.

Figure 1 displays a bar graph depicting the relationship between depression status (PSD, No depression) and stroke type (ischemic, hemorrhagic). There was a higher percentage of patients with an ischemic stroke with no depression (N = 160 out of 325 patients) compared to all other groups (2-sided *p* = 0.022).

Table 4 displays the logistic regression analysis using depression status as the dependent variable revealing five factors associated with a diagnosis of PSD (*p* < 0.001). Specifically, increased odds of PSD were associated with a comorbid anxiety disorder (OR = 5.95, *p* < 0.001); patients with comorbid clinical anxiety were about 6 times more likely to suffer from PSD. Further, increased odds of PSD were associated with a history of depression before stroke (OR = 3.00, *p* < 0.001), experiencing fatigue (OR = 2.75, *p* = 0.002), having memory impairment (OR = 2.36, *p* = 0.019), and younger age at stroke (*p* = 0.006). Time since stroke, chronological age, location of the stroke (e.g., basal ganglia), lateralization of stroke, stroke type (hemorrhagic/ischemic), sleep difficulties, sex, employment status, marital status, anxiety treatment before stroke, and the presence of executive difficulties were not associated with increased odds of PSD.

Though not hypothesis guided, we sought to explore the bidirectional relationship between PSD and poststroke anxiety. To do so, we conducted another logistic regression model using poststroke anxiety disorder as the dependent variable and all of the variables in Table 4 as independent variables. The premise for this was that people with poststroke anxiety may be more likely to suffer from PSD. Results of the logistic regression with poststroke anxiety disorder as the dependent variable indicated the following significant predictors: poststroke depression (OR = 5.8, *p* < 0.001), sleep difficulties (OR = 3.1, *p* = 0.003), and female gender (OR = 2.8, *p* = 0.009).

## 4. Discussion

This study extends the literature by identifying a comorbid DSM-diagnosed anxiety disorder as the strongest factor associated with increased odds of PSD, primarily panic attacks and GAD. PSD was almost 6 times more likely to occur in individuals with a comorbid anxiety disorder. Previous studies have either found an association between anxiety and depression after stroke using screening measures [22] or predicted future PSD by assessing level of anxiety using the Hospital Anxiety and Depression Scale (HADS) [23,23]. Sagen et al. [24] found that significant anxiety symptoms 2 weeks after stroke predicted PSD 4 months after stroke; although, it is important to note that the HADS does not screen for all anxiety disorders. Recently, using Medicare diagnoses of history of depression, anxiety, and stroke (N = 174,901 patients with stroke), Mayman et al. [19] found that history of anxiety was the strongest predictor of being diagnosed with depression within a year and a half after stroke. They also found that being discharged to home after an acute hospitalization after the stroke was a protective factor against PSD. One caveat of the study [19] was its reliance on ICD diagnostic codes identified through Medicare billing that is different than clinical diagnosis of an anxiety or mood disorder. Clinicians from different specialties use different unknown screening measures to diagnose anxiety.

Fatigue and sleep difficulties were more common in patients with PSD than those with no depression, consistent with previous research [25,26]. An increased frequency of fatigue and sleep issues can be found in PSD considering a depression diagnosis is based on these symptoms. Notably, when fatigue and sleep difficulties were entered into the logistic regression together, it was fatigue that was the contributing factor accounting for increased odds of PSD. Fatigue and sleep difficulties are highly correlated, and both can contribute to poststroke depression such that sleep deprivation worsens mood when the body cannot refresh, and when the body feels tired, the mind can have less reserve to fight ruminations and worries [27] further contributing to depression. Importantly, the American Heart Association Council on Cardiovascular and Stroke Nursing and Stroke Council made it a point of emphasis to find effective interventions for fatigue that can be disabling and can limit recovery after stroke [28].

The association between mood and cognition in depressed individuals without a comorbid neurologic disorder is well documented. Depressed individuals are more likely to have impaired cognitive speed, memory, naming ability, planning, and problem-solving compared to non-depressed individuals [29]. The association between depression and cognitive difficulties in individuals with PSD is thought to reflect the effect of co-occurring brain damage (hippocampal damage leads to forgetfulness and emotion-regulation difficulties) or changes in environment (sadness with reduced opportunities leading to trouble with attention and forgetfulness) [30]. Recently, Kang [31] found that those with severe PSD were more forgetful. Indeed, mild levels of depression are not always associated with cognitive deficits, suggesting that the use of screeners to diagnose depression will be misleading when the diagnosis was mild sadness instead of clinical depression [31]. It is interesting to note that in our study there were higher rates of both memory and executive impairment in PSD compared to stroke patients without depression when evaluated in between-group analyses, but when used side by side in the regression analysis, the odds of PSD were significantly associated with memory loss and not executive difficulties.

In our study, individuals who had a history of treatment for depressive symptoms before the stroke were 3 times more likely to suffer from PSD. Others have also found an association between PSD and a history of treatment of psychiatric illness [11]. Using meta-analysis, Shi et al. [18] found that a history of treatment for psychiatric illnesses prior to stroke was a consistent predictor of future depression, especially in the first three months after stroke. They speculated that psychiatric illness is known to be recurrent, therefore, increasing the risk of depression after a traumatic event such as a stroke. They also speculated that there were other factors associated with psychiatric illness yet to be studied that could contribute to PSD in individuals with a history of pre-stroke depression, including neuroticism and a family history of mental disorders. Future studies could look at the quality of PSD in individuals with a history of treatment for psychiatric illness prior to the stroke versus those with first-ever diagnoses of a depressive episode after the stroke.

In this study, a younger age at stroke also predicted PSD. Jaroonpipatkul et al. [32] found that interactions between chronological age, white matter hyperintensity load, and hypertension predicted depression after stroke. It may be that older patients with a reduced cognitive reserve secondary to prestroke white matter disease and hypertension are at greater risk for comorbidities such as psychiatric illness after stroke. Interestingly, while studying the prediction of anxiety after stroke, Kapoor et al. [33] found that the presence of depressive symptoms and younger age were the strongest predictors of generalized anxiety disorder after transient ischemic attack or stroke. They speculated that chronological age may predict anxiety after stroke given the finding of a higher prevalence of anxiety disorders in younger stroke survivors.

There are several mechanisms to explain the relationships among PSD and other variables. As reviewed by Medeiros et al. [2], these include prestroke factors (health, history of psychiatric illness), environmental factors (disability, isolation), stroke factors (left-lateralized, location of stroke more anterior in the brain), and pathophysiology factors (glutamate toxicity, abnormal neurotropic response, lower levels of monoamines, and increased inflammation/HPA axis dysregulation). Related to the current study, a poststroke anxiety disorder and associated cortisol dysregulation from increased stress or decreased ability to manage stress could make a person more susceptible to PSD [34]. Being younger and more anxious may lead to increased social anxiety or agoraphobia that in turn worsens mood. Further, fatigue could limit social interactions and make the mind feel down, worsening mood. Finally, the persistence of memory loss may exacerbate depression because of the reduced opportunities to return to work or return to enjoyable social interactions with others because of losing information quickly.

There were several factors in this study that were not associated with PSD in the regression analysis including time since stroke, lateralization and location of stroke, sex, sleep difficulties, employment status, marital status, and the presence of executive difficulties. Individual demographic differences and other characteristics between study samples, smaller sample sizes, inclusion/exclusion of particular variables, and the use of different tools to define PSD may sometimes explain why variables do not show up as significant predictors. Importantly, most studies have not included DSM diagnosis of an anxiety disorder as a predictor of PSD. Our study suggests that when poststroke anxiety disorders are entered into regression analysis, other common risk factors for depression may become less relevant.

The primary limitation to this study was that it was cross-sectional, not allowing for prediction of future events. Future studies should use a DSM diagnosis of poststroke anxiety disorders in a longitudinal study to understand the future risk of PSD in the years after stroke [33]. Another limitation was not being able to divide groups by the location of stroke (frontal left versus frontal right, versus basal ganglia, given the limited sample size of those subgroups). Previous research has shown that the location of stroke predicts PSD (e.g., left frontal), though this finding remains controversial. Another limitation to consider would be that our average encounter with patients was about 8 months after stroke. Some research demonstrates that significant depression may occur in the first 3 months after stroke that improves as circumstances improve, and some significant depression may show up after a year as functional and social limitations persist. Our study was focused on more short-term depression and may reveal different associations compared to patients with more chronic poststroke depression. Finally, future research could look at the efficacy of a DSM diagnosis of specific anxiety disorders in predicting future poststroke depression. Chun et al. [35] found that phobic disorders (e.g., social anxiety) and generalized anxiety disorder were the most common types of anxiety after stroke.

## 5. Conclusions

Poststroke screening and treatment plans for depression should also address comorbid anxiety, given its prevalence as demonstrated in our findings and the literature. Our study showed that poststroke depression affected one-third of our patients leading to worse functional outcome and increased morbidity. PSD is associated with several factors that may be responsive to interventions, including the management of memory and executive difficulties with cognitive rehabilitation and speech pathology therapy, interventions for fatigue including exercise regimens and medications, interventions for sleep difficulties, and interventions for anxiety disorders such as group support or medication that specifically targets the particular anxiety disorder [36,37]. There is evidence that the treatment of PSD early on after stroke may lead to improved cognition and physical recovery, as well as increased longevity [8,38]. Managing these different factors will likely only help alleviate the burden of depression after stroke.

## Figures and Tables

**Figure 1 ijerph-20-05328-f001:**
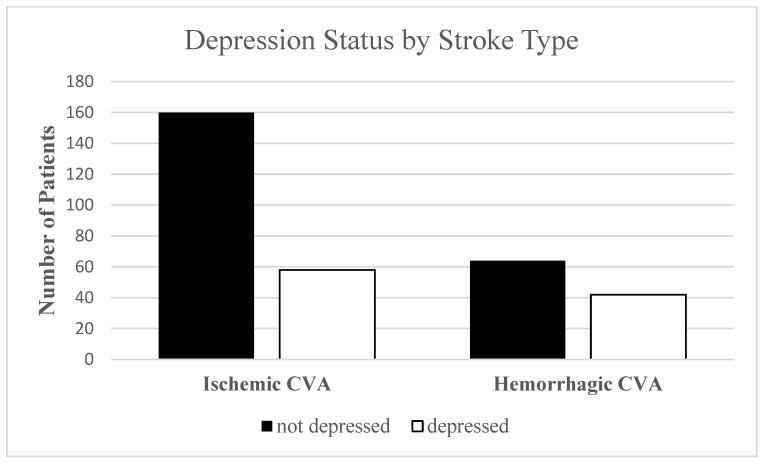
The relationship between depression status (PSD, No depression) and stroke type (ischemic, hemorrhagic). There was a higher number of patients with an ischemic stroke (160 of the 325 patients) with no depression compared to all other groups (2-sided *p* = 0.022).

**Table 1 ijerph-20-05328-t001:** List of Demographic, Stroke, Psychiatric, Psychosocial, and Cognitive Characteristics of the Sample, N = 325.

	%	Mean (SD)
Chronological age (years)		59.3 (13.8)
Education level (years)		14.6 (3.0)
Time since stroke to neuropsych testing (mo.)		8.1 (5.2)
Side of stroke		
left-sided	44.9%	
right-sided	41.2%	
bilateral	13.8%	
Stroke type		
ischemic	67.1%	
hemorrhagic	32.9%	
Sex		
women	49.2%	
men	50.8%	
Poststroke depression	30.8%	
Depression treatment before stroke	38.2%	
Poststroke anxiety	16.9%	
Anxiety treatment before stroke	7.4%	
Sleep difficulties	45.5%	
Fatigue	59.1%	
Not employed	68.9%	
Marital status		
Married/Domestic partner	63.7%	
Unmarried/Divorced/Single	36.3%	
% with memory loss	67.7%	
% with executive difficulties	72.0%	

**Table 2 ijerph-20-05328-t002:** Types of poststroke anxiety disorders in the sample.

No PSA	83.1% (N = 270)
Panic Attacks	6.2% (N = 20)
Generalized Anxiety Disorder (GAD)	4.9% (N = 16)
Obsessive Compulsive Disorder	0.0% (N = 0)
Post-traumatic Stress Disorder (PTSD)	0.6% (N = 2)
Social Anxiety, phobia	1.5% (N = 5)
Anxiety NOS	0.6% (N = 2)
Panic Attacks and GAD	1.8% (N= 6)
GAD and Social Anxiety	0.9% (N = 3)
PTSD and Social Anxiety	0.3% (N = 1)

**Table 3 ijerph-20-05328-t003:** Display of demographic, stroke-related, psychosocial, and cognitive characteristics divided by depression status (poststroke depression, no depression).

	PSD (N = 100)	No Depression (N = 215)	
	% (N)	% (N)	Statistics
Chronological age (years)	55.7 (14.4)	60.8 (13.2)	F(1,323) = 10.0, *p* = 0.002 *
Education level (years)	14.3 (2.6)	14.8 (3.2)	F(1,323) = 2.11, *p* = 0.147
Time since stroke to neuropsych testing (mo.)	9.7 (5.8)	7.4 (6.8)	F(1,323) = 1.25, *p* = 0.261
Side of stroke			
left-sided	45% of depressed	45% of depressed	chi-square *p* = 0.72
right-sided	42.2%	39.0%	
bilateral	12.8%	16.0%	
Stroke type			
ischemic	58.0%	71.1%	*p* = 0.015 *
hemorrhagic	42.0%	28.9%	
Sex			
women	55.0%	46.7%	*p* = 0.103
men	45.0%	53.3%	
Depression treatment before stroke	61.0%	28.0%	*p* < 0.001 *
Poststroke anxiety	40.0%	9.3%	*p* < 0.001 *
Anxiety treatment before stroke	20.0%	2.0%	*p* < 0.001 *
Sleep difficulties	63.0%	37.0%	*p* < 0.001 *
Fatigue	79.0%	50.2%	*p* < 0.001 *
Not employed	62.0%	65.3%	*p* = 0.32
Married/Domestic partner	62.0%	64.4%	*p* = 0.46
% with memory loss	79.0%	62.7%	*p* = 0.02 *
% with executive difficulties	79.0%	68.8%	*p* = 0.39 *

* reflects between group differences.

**Table 4 ijerph-20-05328-t004:** Results of the logistic regression analysis indicating poststroke anxiety, history of depression treatment before stroke, fatigue, memory impairment, and younger age were significant associated factors of PSD.

Significant Factors	OR	*p*-Value	95% CI	
Poststroke anxiety	5.95	0.00	2.42	14.61
Depression treatment before stroke	3.00	0.00	1.62	5.55
Fatigue	2.75	0.00	1.36	5.58
Memory impairment	2.36	0.02	1.16	4.80
Chronological age (years)	0.97	0.01	0.94	0.99
**Non-significant factors**				
Time since stroke to neuropsych testing (mo.)	1.00	0.69	0.99	1.02
Chronological age (years)	0.97	0.01	0.94	0.99
Location of the stroke	1.30	0.43	0.68	2.47
Side of stroke	0.93	0.72	0.61	1.41
Stroke type	1.65	0.11	0.90	3.03
Sleep difficulties	1.45	0.25	0.77	2.76
Employment status	0.65	0.24	0.32	1.32
Marital status	1.01	0.93	0.80	1.27
Anxiety treatment before stroke	1.43	0.63	0.34	6.04
Executive impairment	1.30	0.47	0.63	2.68

## Data Availability

Data are available for verification upon request.

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
