# Peer review of "Biological, Psychiatric, Psychosocial, and Cognitive Factors of Poststroke Depression"

_ijerph, 2023, doi:10.3390/ijerph20075328_

Round 1

Reviewer 1 Report

Thank you for submitting this interesting paper. Overall I believe that it will be a very useful addition to the literature. I have made a few suggestions on the manuscript that you may wish to address.

Author Response

Reviewer #1 comments

  1. General comment: I have made a few suggestions on the manuscript that you may wish to address.

Author reply: Thank you. We made all recommended changes to the manuscript. Changes were highlighted.

Regarding the following comments:

  1. Methods section: I would suggest rewriting this paragraph as inclusion/exclusion criteria, possibly in tabular form.

Author reply: We crafted a bullet point inclusion/exclusion section to make the information clearer.

  1. Statistical analysis section: Reviewer 1: wondering if this would look better in a table?

Author reply: Similarly, we crafted a bullet point inclusion/exclusion section to make the information clearer.

Reviewer 2 Report

The title is not a good  representative of aspects of the study.it is only present one novelty of the work.

this is the study in the adults patients not included the youth that should be emphasised in the title.

methodologi is fair however the assessment of fatigue is completely via interview that could not be perfect.

Mixed anxiety detection is novel,it was better to determine the types.

the analysis of the locations of stroke and relationship to the risks is important that it has not been performed.

The presentation of the results should be shorter,preferentially by tables and figures.

Discussion is also reppeats in some similar aspects,i assume it must be more organized.

Author Response

Reviewer #2 comments

1. Title page: The title is not a good representation of aspects of the study. It is only presenting one novelty of the work.

Author reply: We have made the title more generic to better reflect the study.

2. Title page: This is the study in the adults patients did not include the youth that should be emphasized in the title.

Author reply: The title now reflects the fact that this study did not focus on a pediatric population

3. Methods (page 6): Methodology is fair. However, the assessment of fatigue is completely via interview that could not be perfect.

Author reply: We agree that ‘fatigue’ can be a subjective and ambiguous term. While issues with fatigue were initially asked during the clinical interview, we also checked the referring physicians medical notes to assess if there were complaints of fatigue. Also, the Beck Depression Inventory-II (BDI-II) was used confirm if the patient was suffering from fatigue. The BDI-II has 2 sections that estimate level of ‘Loss of Energy’ and level of ‘Tiredness or Fatigue’ The patient chooses from 4 options such as no problem or very severe. The following sentence is in the Methods to help readers know that fatigue was estimated in several ways. “Severity of fatigue or tiredness was estimated based on the Beck Depression Inventory-2 questions asking about tiredness or fatigue. “

4. Mixed anxiety detection is novel. It was better to determine the types.

Author reply: We have added the subtypes types of anxiety disorders to the RESULS section and TABLE 2 now reflects the types of poststroke anxiety disorders. Also, history of an anxiety disorder was added to the logistic regression analysis.

5. The analysis of the locations of stroke and relationship to the risks is important and has not been performed.

Author reply: Localization of stroke and side of stroke were added to the logistic regression analyses. All sections of the paper were updated regarding changes in values for the analyses. Notably, while the ORs changed slightly, the pattern of prediction did not change, location of stroke and side of stroke did not change the prediction of poststroke depression. Likely, location did not change as a predictor because our very diverse group of stroke locations (35.7% MCA, 23.1% frontal lobe/ACA stroke, 14.8% cerebellar, 9.9% basal ganglia, 6.5% 4.6% pontine, thalamic, 3.4% PCA, 3.4% occipital, 1.8% parietal.).

6. The presentation of the results should be shorter, preferentially by tables and figures.

Author reply: Results section was amended to hopefully make it more linear, and Tables and a Figure were added to help provide more visual depictions.

7. Discussion also repeats in some similar aspects. I assume it must be more organized.

Author reply: Very good. Yes, given the request for lengthening the paper by the journal, we added more length initially that led to this redundancy. We’ve worked to reduced redundancy. Each paragraph was created to focus specifically on each of the specific predictors of poststroke depression.

Reviewer 3 Report

Dulay et al. report on a prospectively recruited cohort of 335 stroke patients of which 31% developed post-stroke depression. Overall, the study design is skillfully executed with well-defined endpoints and sensible conclusions. A particular strength of the manuscript is the comprehensive review of prior literature relating to the current study results. But before publication there is a number of minor and major points, which I would kindly ask the authors to address.

Minor: If PSD commonly develops during the first year after stroke, is it a limitation that patients were evaluated <12months after stroke (mean 7.5 months).

Minor: Were there differences between hemorrhagic and ischemic stroke patients in regard to depression and anxiety?

Minor: Were patients excluded based on their inability to attend follow-up examinations at the neuropsychology department or were immobile patients or patients living in nursing homes after stroke included as well?

Major: It does not become clear whether comorbid anxiety disorder might have been preexisting or developed after the index event. Are there any information on presence of anxiety before stroke?

Major: The strong association of anxiety disorder with PSD seems to be mainly explained by the low incidence in patients without depression. Consequentially, could depression be a stronger predictor for anxiety than vice versa? Therefore, it seems more plausible to interpret the results the other way round, i.e. that patients with PSD are more likely to suffer from poststroke anxiety and give the odds ratio for anxiety as a dependent variable. Nonetheless, this would keep the conclusion of the study intact, namely that comorbid anxiety needs to be addressed as a relevant, disease-sustaining factor of PSD.

Author Response

Reviewer #3 comments

1. Minor: If PSD commonly develops during the first year after stroke, is it a limitation that patients were evaluated <12months after stroke (mean 7.5 months).

Author reply: Good point, there are likely different etiologies for depression for ‘early onset’ PSD and ‘late onset’ PSD. Some studies show significant depression in the first 3 months after stroke that may improve as circumstances improve and some significant depression may show up after a year as functional and social limitations persist. We have added a caveat to the limitations section to address your point that patients were evaluated before a year after stroke on average: “Another limitation to consider would be that our average encounter with patients was about 8 months after stroke. Some research demonstrates that significant depression may occur in the first 3 months after stroke that improves as circumstances improve and some significant depression may show up after a year as functional and social limitations persist. Our study then was focused on more short-term depression and may reveal different associations compared to patients with more chronic poststroke depression.”

2. Minor: Were there differences between hemorrhagic and ischemic stroke patients in regard to depression and anxiety?

Author reply: Additional information and analyses were conducted to evaluate this question. The following was added to the METHODS section: “There was a higher percentage of patients with an ischemic stroke with no depression compared to all other groups (2-sided p=0.022).” Figure 1 was created. Also, stroke type was added to the logistic regression analyses. We found that 10 patients in our sample had an indeterminate stroke type so we chose to exclude them from the study. Note: removing those 10 patients (patients who had longer time since stroke with missing confirmative data) made the time since CVA jump from 7.5 months to 8.1 months.

3. Minor: Were patients excluded based on their inability to attend follow-up examinations at the neuropsychology department or were immobile patients or patients living in nursing homes after stroke included as well?

Author reply: Patients had to make it to the hospital. The facility is very accommodating for immobile patients. No one was excluded based on immobility or inability to attend. We added the following to the METHODS section for further clarification: “Evaluations only occurred at the Houston Methodist hospital in quiet dedicated clinic rooms. Patients came from diverse places including home, retirement communities, inpatient rehabilitation facilities, and skilled-nursing facilities.” and “No patients were excluded because of immobility. If patients could not use their upper extremities to write, non-motor neurocognitive tests were used to define memory of executive impairments.”

4. Major: It does not become clear whether comorbid anxiety disorder might have been preexisting or developed after the index event. Is there any information on presence of anxiety before stroke?

Author reply: More detail about history of treatment for an anxiety disorder prestroke has been added to the RESULTS section. The following was added to the 1st paragraph of the RESULTS section, “of the 55 patient in this study diagnosed with a poststroke anxiety disorder, 43.6% (N=24) of patients had been treated for an anxiety disorder prior to the stroke, and in nearly every case the anxiety type was the same (e.g., the 2 patients with PTSD had re-occurrence of their longstanding PTSD after the strokes). New onset poststroke anxiety occurred in 56.4% of the 55 patients (N=31).

5. Major: The strong association of anxiety disorder with PSD seems to be mainly explained by the low incidence in patients without depression. Consequentially, could depression be a stronger predictor for anxiety than vice versa? Therefore, it seems more plausible to interpret the results the other way round, i.e., that patients with PSD are more likely to suffer from poststroke anxiety and give the odds ratio for anxiety as a dependent variable. Nonetheless, this would keep the conclusion of the study intact, namely that comorbid anxiety needs to be addressed as a relevant, disease-sustaining factor of PSD.

Author reply: This is an excellent point. We chose PSD as a dependent variable because depression after stroke occurs at a higher frequency. It’s likely absolutely a bidirectional relationship, maybe often labeled anxious depression. Sometimes people who are depressed are also anxious, and sometimes people who are anxious can be depressed. For novelty, we could have used poststroke anxiety as the dependent measure at the outset of this prospective study. Our initial hypotheses before starting this project though focused on PSD.  If ok, we’d prefer to keep the study focused on PSD, but we’ve added a small section to the end of the RESULTS section to address your insightful observation: “Though not hypothesis guided, we sought to explore the bidirectional relationship between PSD and poststroke anxiety. To do so, we conducted another logistic regression model using poststroke anxiety disorder as the dependent variable and all of the variables in Table 4 as independent variables. The premise for this was that people with poststroke anxiety may be more likely to suffer from PSD. Results of the logistic regression with poststroke anxiety disorder as the dependent variable indicated the following significant predictors: poststroke depression (OR=5.8, p<0.001), sleep difficulties (OR=3.1, p=0.003), and female gender (OR=2.8, p=0.009).”

Round 2

Reviewer 2 Report

Thanks to perform all the requested changes